# Both electronic and vibrational coherences are involved in primary electron transfer in bacterial reaction center

Fei Ma [1], Elisabet Romero[1], Michael R. Jones[2], Vladimir I. Novoderezhkin[3] & Rienk van Grondelle[1]

Understanding the mechanism behind the near-unity efficiency of primary electron transfer in reaction centers is essential for designing performance-enhanced artificial solar conversion systems to fulfill mankind's growing demands for energy. One of the most important challenges is distinguishing electronic and vibrational coherence and establishing their respective roles during charge separation. In this work we apply two-dimensional electronic spectroscopy to three structurally-modified reaction centers from the purple bacterium *Rhodobacter sphaeroides* with different primary electron transfer rates. By comparing dynamics and quantum beats, we reveal that an electronic coherence with dephasing lifetime of ~190 fs connects the initial excited state, $P^*$, and the charge-transfer intermediate $P_A^+ P_B^-$; this $P^* \rightarrow P_A^+ P_B^-$ step is associated with a long-lived quasi-resonant vibrational coherence; and another vibrational coherence is associated with stabilizing the primary photoproduct, $P^+ B_A^-$. The results show that both electronic and vibrational coherences are involved in primary electron transfer process and they correlate with the super-high efficiency.

[1] Department of Biophysics, Faculty of Sciences, VU University Amsterdam, De Boelelaan 1081, 1081 HV Amsterdam, The Netherlands. [2] School of Biochemistry, University of Bristol, Biomedical Sciences Building, University Walk, Bristol BS8 1TD, UK. [3] A. N. Belozersky Institute of Physico-Chemical Biology, Moscow State University, Leninskie Gory, Moscow 119992, Russia. Correspondence and requests for materials should be addressed to F.M. (email: fma@iccas.ac.cn)

The idea of quantum coherence playing a role in photo-synthesis arose from observations that some energy or electron transfer processes in bacterial and plant pigment–protein complexes are efficient to an extent that exceeds explanation using only classical theory[1,2]. Since its first implementation[3], two-dimensional electronic spectroscopy (2DES) has become a powerful tool for the study of coherent mechanisms in photosynthetic complexes, with broadband excitation creating coherent superpositions of electronic/vibrational states that give rise to specific features in the 2D spectra such as quantum beats (QBs). Explanations for observed long-lived QBs have evolved during the past decade. While electronic coherence was first proposed as the origin[4], more and more experimental and theoretical evidence has supported proposals that long-lived QBs arise from vibrational[5] or vibronic (electronic–vibrational mixed)[6–11] coherence. Recently, discrimination of diverse coherences showed that electronic coherence completes in <200 fs while vibrational/vibronic coherence can persist for as long as 2 ps[12].

Reaction centers (RCs) facilitate the key reaction of photosynthesis in which harvested solar energy is converted into a trans-membrane electrochemical potential with a near-unity quantum efficiency (charges separated per photon absorbed). Light-powered charge separation is accomplished by (bacterio)

chlorin cofactors embedded in a protein matrix. In the RC of the purple bacterium *Rhodobacter* (*Rba*.) *sphaeroides*, the cofactors are a pair of bacteriochlorophyll (BChl) *a* (P), two monomeric BChl *a* (B) and two bacteriopheophytin (BPhe) *a* (H) molecules arranged in two branches around an axis of quasi two-fold symmetry (Fig. 1a)[13]. The P BChl pair, $B_A/B_B$ BChls, and $H_A/H_B$ BPhes exhibit near-discrete absorption bands (Fig. 1b). Charge separation is initiated from the excited state of P (P*), forming a partial intradimer charge-transfer (CT) intermediate, $P_A^+P_B^-$. Subsequently an electron is transferred to $B_A$, $H_A$, and a ubi-quinone ($Q_A$) sequentially, which make up the "active branch" or "A-branch." At 77 K, the electron transfer (ET) processes $P^* \rightarrow P^+B_A^-$, $P^+B_A^- \rightarrow P^+H_A^-$, and $P^+H_A^- \rightarrow P^+Q_A^-$ occur with time constants of 1−2 ps, 0.5−1 ps, and ~200 ps, respectively[13,14]. The coherent energy transfer in *Rba. sphaeroides* RC was extensively studied with 2DES[11,15–18], where in most cases P was oxidized to $P^+$ and cannot perform ET.

In 1993, coherent nuclear motion affecting P* was visualized and was suggested to play a functional role in the primary ET process, $P^*(P_A^+P_B^-) \rightarrow P^+B_A^-$[19]. Subsequent theoretical models showed that coupling of these nuclear vibrational modes facilitates this process by removing the barrier for forming $P_A^+P_B^-$ and stabilizing $P^+B_A^-$[20–23]. In 2014, 2DES studies on the related

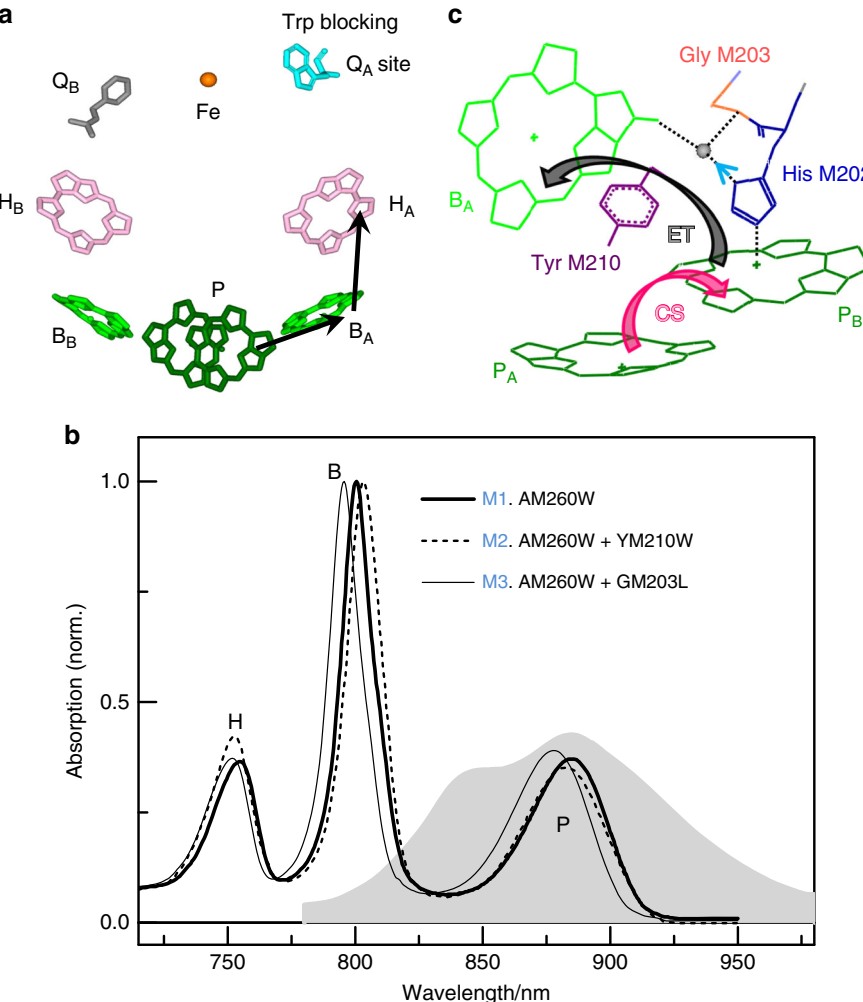

**Fig. 1** Cofactor structure and absorption spectra of reaction centers (RCs). **a** Arrangement of cofactors and the electron transfer (ET) pathway in **M1**, which lacks a $Q_A$ acceptor due to an alanine to tryptophan replacement (the crystal structure is from PDB record 1QOV[20]). **b** Normalized absorption spectra of **M1** (solid line), **M2** (dashed line), and **M3** (light solid line) at 77 K overlaid with the laser spectrum (gray shaded area). **c** Scheme of charge separation (pink arrow) and ET (black arrow) with mutation site for **M2** (Tyr M210, purple) and **M3** (Gly M203, red). A water molecule (gray ball) links $P_B$ and $B_A$ via a hydrogen-bond interaction involving the former's axial histidine M202 (dotted lines)

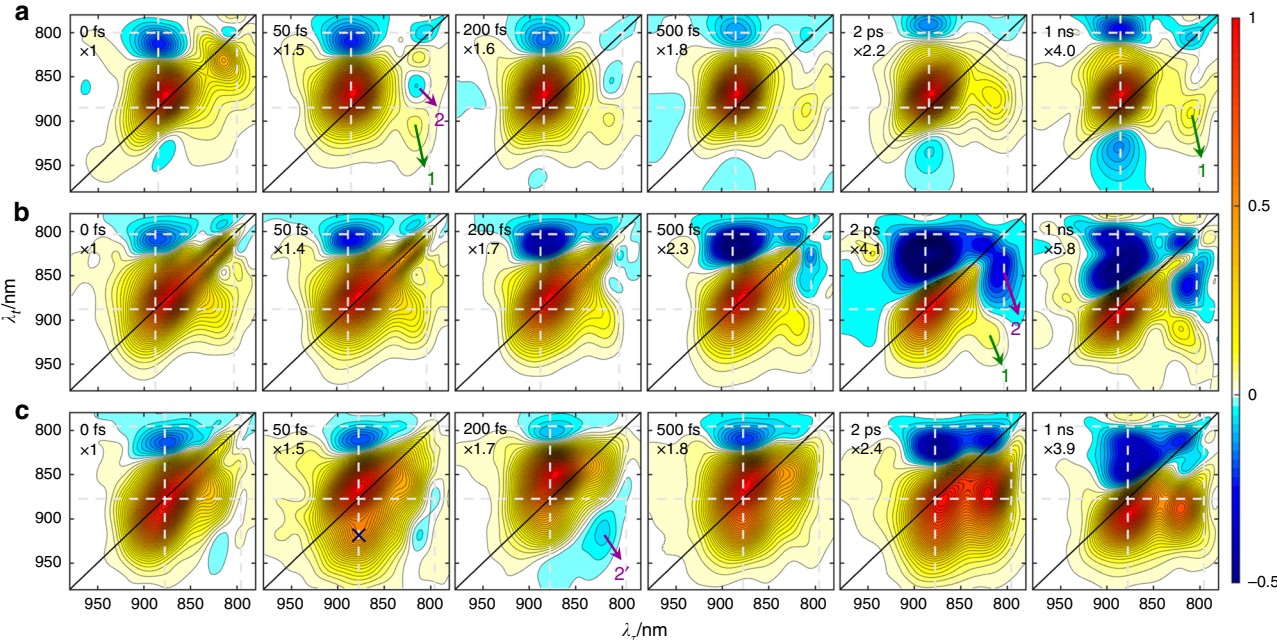

**Fig. 2** 77 K absorptive total two-dimensional (2D) spectra. 2D spectra of **M1** (**a**), **M2** (**b**), and **M3** (**c**). Symbols $\lambda_\tau$ and $\lambda_t$ denote the excitation and detection wavelengths, respectively; the indicated population time $T$ is shown in the left-top corner of each panel. The spectra are normalized to the maximum of the diagonal P signal; the relative amplitude multiplier is shown below the $T$

Photosystem II (PSII) RC from oxygenic phototrophs exhibited that coherence between exciton and CT states correlates with the efficient and ultrafast charge separation[8,9]. However, owing to the high spectral congestion, it was difficult to distinguish electronic and vibrational coherence. To address this issue, we recently applied room-temperature 2DES to a structurally altered *Rba. sphaeroides* RC in which an alanine at position 260 of the M-polypeptide is replaced by tryptophan (AM260W)[24]. This RC assembles without a $Q_A$ ubiquinone, guaranteeing that no inactive RCs with an oxidized P persist beyond the repetition time between excitations during 2DES (1 ms in this work). At the same time, the $P^* \rightarrow P^+B_A^- \rightarrow P^+H_A^-$ ET rates are essentially identical to those in the wild-type RC[25,26]. From 2DES of this RC, we identified long-lived excited-state vibrational modes that may couple with the ET process.

To distinguish electronic and vibrational coherence and identify their respect roles, in this work we extend our previous study by analyzing two double-site mutant RCs that exhibit slowed primary ET rates in addition to the $Q_A$-excluding AM260W mutation. 2DES experiments are conducted at 77 K rather than the room temperature used in our previous work, producing more distinguishable spectral shapes and thus revealing short-lived intermediates that are usually hidden in a room-temperature experiment. Clear correlations are found (1) between electronic coherence and the rate of ultrafast $P^* \rightarrow P_A^+P_B^-$ charge separation, and (2) between long-lived vibrational coherence and the rate of the slower $P^*(P_A^+P_B^-) \rightarrow P^+B_A^-$ ET. The results show that both types of coherence are involved in optimizing primary ET.

## Results

**Structure and absorption of the mutant RCs**. The AM260W mutation was combined with replacement of the tyrosine at position 210 of the M-polypeptide with tryptophan (YM210W) or glycine at position 203 of the M-polypeptide with leucine (GM203L) (Fig. 1c). The AM260W mutation does not change the structure of $P_A$, $P_B$, and $B_A$ or their protein surroundings[25]. The YM210W mutation causes a small tilt of the macrocycle of $B_A$ and produces a 50-mV increase in the $P^+/P$ mid-point redox potential,

with the consequence that primary ET is slowed by as much as two orders of magnitude[27,28]. The GM203L mutation removes a crystallographically resolved water molecule, which directly links $P_B$ and $B_A$ by hydrogen-bond interactions with the oxygen of the 13$^1$-keto carbonyl group of $B_A$ and the nitrogen of the histidine (M202) that provides the axial ligand to the magnesium of $P_B$[29]. This results in a one-order slowing of the primary ET rate[30,31]. For convenience, the AM260W, AM260W+YM210W, and AM260W+GM203L mutant RCs are termed as **M1**, **M2**, and **M3** in this report.

Both the positions and relative amplitudes of the main $Q_y$ absorption bands change between the three RCs. At 77 K, the P/B bands peak at 884.3/800.5, 882.5/803, and 878/795.5 nm for **M1**, **M2**, and **M3**, respectively (Fig. 1b). It is notable that the position of the P peak relative to that of the B peak is more blue-shifted in **M2** than in **M1** and **M3**. Stark spectroscopy has revealed an internal CT $P_A^+P_B^-$ character in $P^*$, which is the lowest energy level of the strongly coupled P pair[32]. This CT state resides in the red tail of the P band and is responsible for a ~4 nm red shift of the P band. Hence, the blue shift of P may reflect less CT character in **M2**.

**Two-dimensional electronic spectroscopy**. 2DES reveals coherent effects and energy/ET pathways using three ultrashort and spectrally broad laser pulses separated by controlled time delays. Fourier transform (FT) with respect to the coherence time $\tau$ (time between the first and second pulses) and with respect to the rephasing time $t$ (time between the third pulse and the signal) yields the 2DES in the frequency/wavelength domain that correlates the absorption, $\omega_\tau$, and emission, $\omega_t$, frequencies for a fixed population time $T$ (time between the second and third pulses)[3]. 2DES was applied to purified **M1**, **M2**, and **M3** RCs at 77 K. The excitation pulse predominantly excited P with some minor excitation of B (Fig. 1b). With this excitation, the signals around the B absorption wavelength were able to be probed while fast $B^* \rightarrow P$ energy transfer (200 fs[33]) was negligible compared to the $P^*$-initiated ET pathway.

The absorptive 2D spectra at different population times are shown in Fig. 2. At $T = 0$ fs, positive ground state bleach (GSB)

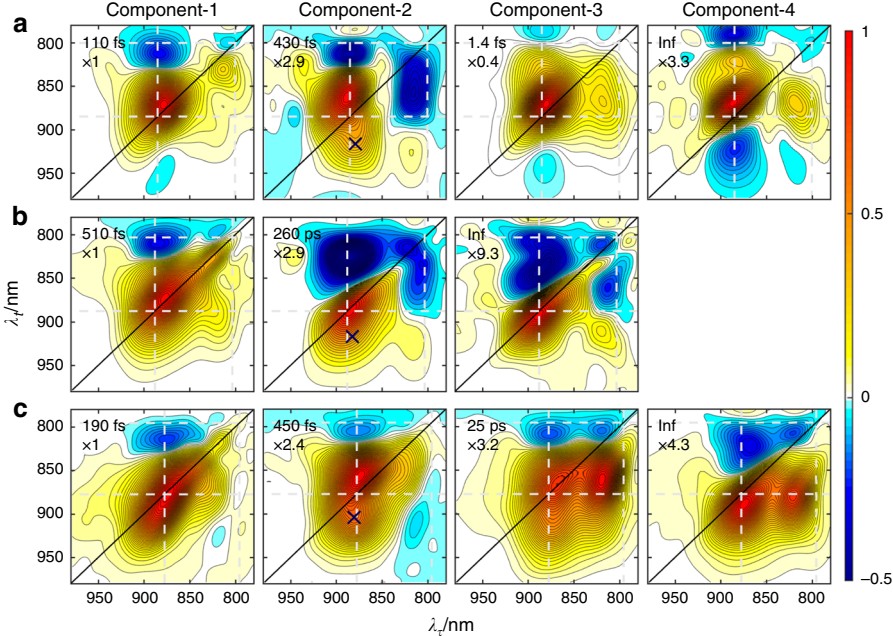

**Fig. 3** Two-dimensional evolution-associated spectra (2D-EAS). 2D-EAS of **M1** (**a**), **M2** (**b**), and **M3** (**c**). The time constants of each species are shown in the left-top corner of each panel. The spectra are normalized to the maximum of the diagonal P signal; the relative amplitude multiplier is shown below the time constants. The crosses in each component-2 EAS mark the approximate location of the $\left(P^*, P_A^+ P_B^-\right)$ cross peak

and stimulated emission signals on the diagonal and the negative excited-state absorption (ESA) signals off the diagonal appeared, associated with the P → P* (major) or B → B* (minor) transition, can be distinguished. The peak amplitude of B GSB relative to that of P had the relationship: **M2 > M1 > M3**. This tendency is consistent with the amount of B* relative to P* changing due to spectral shifts caused by the mutation. A small cross peak centered at around $(\lambda_\tau, \lambda_t) = (815, 880)$ nm was visible. It may reflect dynamic coherence between P* and B* at $T = 0$ fs. The subsequent evolution of the 2DES with $T$ were different between the three mutant RCs.

**M1 evolution**. For **M1** (Fig. 2a), at $T = 50$ fs the B GSB decreased drastically while the P GSB spread along the anti-diagonal direction and a complex signal appeared with four peaks along the $\lambda_\tau = 820$ nm vertical line. The negative (820, 800) nm peak existed from 0 fs until 1 ns and did not shift. The positive (820, 910) nm peak (labeled 1 in Fig. 2) arose from the (B, P) cross peak at $T = 0$ fs and persisted but shifted to (820, 890) nm at $T = 1$ ns. The other two peaks (one positive at (820, 825) nm and the other negative at (820, 865) nm) were short-lived and had nearly disappeared at $T = 500$ fs. The disappearance of the negative (820, 865) nm peak (labeled 2 in Fig. 2) was the reason for the blue shift of the (820, 910) nm peak. The attribution of these three peaks will be discussed in the "Two-dimensional evolution-associated spectra" section below. From 200 fs, another negative off-diagonal peak rose below the P GSB with a center at (885, 925) nm. Meanwhile, the initial negative peak attributable to P ESA at (885, 815) nm blue shifted along the $\lambda_t$ coordinate to (885, 800) nm. As we have demonstrated previously[24,34], these features corresponded to the formation of the $P^+H_A^-$ CT state.

**M2 evolution**. For **M2** (Fig. 2b), the off-diagonal peaks labeled 1 and 2 also appeared. However, completely different from **M1**, the peak labeled 2 grew and remained until 1 ns. The peaks corresponding to the absorption of the $P^+H_A^-$ state did not appear and

the transition of the initial P ESA peak to a red-shifted position (along the $\lambda_t$ coordinate) was still visible at $T = 1$ ns.

**M3 evolution**. For **M3** (Fig. 2c), at $T = 50$ fs, the P GSB signal split with a shoulder peak appearing around (880, 910) nm (cross in Fig. 2c). This splitting was also observed in transient absorption spectra, both with broadband (Supplementary Fig. 1) or single-wavelength[30] excitation. Meanwhile a negative cross peak appeared at (820, 915) nm. This was as short-lived as the peak labeled 2 in **M1** but was 50 nm red-shifted along the $\lambda_t$ coordinate, and so we labeled it as 2'. One should note that from $T = 2$ ps, the (B, P) cross peak labeled 1 was much stronger than those in **M1** and **M2**. Until $T = 1$ ns, no features corresponding to $P^+H_A^-$ were observable, and transition of the P ESA peak to a redshifted position was also visible at $T = 1$ ns.

**Two-dimensional evolution-associated spectra (EAS)**. To obtain the spectral features and evolution time constants of each species, a global analysis was applied. For this multi-step process a sequential model is most suitable because the EAS deviate less from the actual spectra of involved species than for other models such as a parallel model[24,35].

**M1 EAS**. For **M1** (Fig. 3a), four components were extracted with exponential time constants of 110 fs, 430 fs, 1.4 ps, and infinity (the fitting residuals are shown in Supplementary Fig. 2). These values were consistent with previously reported time constants for the respective steps, i.e., ~180 fs for the formation of $P_A^+P_B^-$[36], 1.4 ps for the $P^* \rightarrow P^+B_A^-$ primary ET[13,34], ~0.5 ps for the $P^+B_A^- \rightarrow P^+H_A^-$ secondary ET[13], and ~17 ns for the ground state recovery of $P^+H_A^-$[37]. Hence, the four components should bear the characteristic features of these CT states[24]. Component-1 contained mainly the GSB and ESA signals of the initial excited species, P* and minor B*. Component-2 contained complex structures besides the GSB and ESA of P: (1) a shoulder peak around (880, 910) nm that appeared below the P GSB, (2) a strong negative cross peak around (815, 850) nm. The shoulder peak

corresponded to the GSB of the internal CT state, $P_A^+P_B^-$, whose absorption resides at the red tail of the absorption band of P. The (815, 850) nm cross peak can be attributed to a dynamic Stark shift[18,38]. The Stark responses were the result of the change of the electric field surrounding P and B due to $P^* \rightarrow P_A^+P_B^-$ charge separation. This fingerprint feature, especially the red shoulder of P GSB, can be used for identifying the $P_A^+P_B^-$ CT state.

Component-3 contained two representative features. The main one was the pair of positive cross peaks that were symmetrical to the diagonal, being a strong peak at the (B, P) position and a weaker peak at the (P, B) position. We have concluded in our previous report that a similar feature of **M1** at room temperature corresponded to ET from P to B[24]. Furthermore, this process should be reversible, otherwise the (B, P) peak would not appear (a similar reversible process has been displayed in ref. [39]). Hence, this pair of positive cross peaks represented the $P_A^+P_B^- \rightleftharpoons P^+B_A^-$ ET process with an overall rate of $(1.4\,ps)^{-1}$. The other feature representing $P^+H_A^-$ were the two negative cross peaks, a newly formed (885, 980) nm peak and a (885, 780) nm peak that originated from a blue shift of the P ESA along the $\lambda_t$ coordinate. These cross peaks may also originate from a Stark absorption shift, which was induced by the change of electric field when $P^+H_A^-$ was formed.

The non-decaying Component-4 contained similar features as Component-3 but with much stronger signals representing $P^+H_A^-$. The formation of $P^+H_A^-$ first formed with a time constant of 430 fs and reached a maximum in 1.4 ps. It implied that the $P^+B_A^- \rightarrow P^+H_A^-$ step was faster than the preceding $P_A^+P_B^- \rightleftharpoons P^+B_A^-$ step. Thus the 430 fs and 1.4 ps should correspond to the $P^+B_A^- \rightarrow P^+H_A^-$ and the $P_A^+P_B^- \rightleftharpoons P^+B_A^-$ steps, respectively. The overall process was $P^* \xrightarrow{110\,fs} P_A^+P_B^- \underset{}{\overset{1.4\,ps}{\rightleftharpoons}} P^+B_A^- \xrightarrow{430\,fs} P^+H_A^-$.

**M2 EAS**. Only three components were extracted for **M2** (Fig. 3b), with time constants of 510 fs, 260 ps, and infinity. The data could be very well fitted without the ultrafast 110 fs component used for **M1** (the fitting including this ultrafast component, is shown in Supplementary Note 1 and Supplementary Fig. 3 for comparison). Component-1 corresponded to $P^*$ and a minority of $B^*$. Its lifetime, 510 fs, was somewhat longer than the corresponding lifetime for **M1**. No red shoulder of P GSB corresponding to the $P_A^+P_B^-$ CT state appeared, implying that the initial charge separation process was different from the case in **M1**. Furthermore, the absence of $P_A^+P_B^-$ was consistent with our finding from the steady-state absorption spectra that **M2** possesses less CT character. Component-2, however, contained a Stark absorption shift signal comprising a negative cross peak at (810, 850) nm. It indicated that Component-2 reported a long-lived (260 ps) electric field influence on B. The non-decaying Component-3 contained similar features to Component-2, except that the P ESA peak migrated to a red-shifted position.

The dynamic evolution seen with **M2** indicated that $P^*$ transferred an electron to B with a time constant of 260 ps without forming the $P_A^+P_B^-$ intermediate. The $P^+B_A^- \rightarrow P^+H_A^-$ reaction then occurred with a time constant of 510 fs. The overall process was $P^* \underset{}{\overset{260\,ps}{\rightleftharpoons}} P^+B_A^- \xrightarrow{510\,fs} P^+H_A^-$.

**M3 EAS**. For **M3** (Fig. 3c), four components were extracted with time constants of 190 fs, 450 fs, 25 ps, and infinity. Component-1 represented $P^*$ and a minor population of $B^*$. Component-2 contained the red shoulder of P GSB and thus was attributable to $P_A^+P_B^-$. It also contained a negative cross peak corresponding to the dynamic Stark shift of B, which was however weaker and red-shifted by about 50 nm along the $\lambda_t$ coordinate, compared with

that in **M1**. The extent of this Stark shift reflects the effective electric field strength at a particular cofactor, thus it may imply that, in **M3**, $P_A^+P_B^-$ exerted a stronger electric field strength on B. However, the main reason for the 50 nm difference should be the appearance of the positive (B, P) cross peak, which was much stronger in **M3** and partially obscured the negative Stark absorption signal of B. The formation time of $P_A^+P_B^-$, 190 fs, was longer than the 110 fs for **M1**.

From Component-3 to Component-4, the main change was the migration of the P ESA peak to a red-shifted position. This phenomenon appeared in both **M2** and **M3**. A notable feature in the Component-3 and Component-4 spectra was the positive cross peak at the (B, P) position, whose amplitude was much stronger than in the case of either **M1** or **M2**.

The dynamic evolution seen with **M3** determined that $P_A^+P_B^-$ and $P^+B_A^-$ were formed sequentially with time constants of 190 fs and 25 ps, respectively. The subsequent $P^+B_A^- \rightarrow P^+H_A^-$ step had a time constant of 450 fs. The overall process was $P^* \xrightarrow{190\,fs} P_A^+P_B^- \underset{25\,ps}{\overset{}{\rightleftharpoons}} P^+B_A^- \xrightarrow{450\,fs} P^+H_A^-$. With these rates, $P^+H_A^-$ should have formed in an observable amount at 1 ns; however, what we observed was contrary to this. Considering the strong cross peak at the (B, P) position, the reason for the absence of a $P^+H_A^-$ signal can be explained by increased backward ET from $P^+B_A^-$ to $P_A^+P_B^-$, i.e., charge recombination. This explanation can be validated by comparing the spectral shapes at 77 K and room temperature (Supplementary Note 2 and Supplementary Fig. 4): at room temperature, where the forward ET was slower and less effective, this (B, P) cross peak became stronger. Furthermore, we calculated a kinetic model with reversible reactions and the results showed that, when the backward reaction became more effective, the amplitude of the corresponding cross peak increased (details are given in Supplementary Note 3 and Supplementary Fig. 5).

**The effects of mutation on ET dynamics**. Charge separation in **M1** was very similar to that in the wild-type RC[13], with $P^+H_A^-$ being formed in <1 ps. Charge separation became much less effective upon additional mutating at the M210 or M203 site, with $P^+H_A^-$ not being formed even up to 1 ns. The 2D-EAS provided detailed information on the ET dynamics that led to this degradation of RC function.

Comparing the three mutant RCs, the time constants for the $P^+B_A^- \rightarrow P^+H_A^-$ step were similar, and so the main differences concerned the rates and efficiencies of forming $P^+B_A^-$. In **M1**, the ultrafast charge separation (110 fs for $P_A^+P_B^-$ formation) and relatively fast ET from $P_B$ to $B_A$ (1.4 ps), with noneffective charge recombination, were responsible for the high efficiency of $P^+B_A^-$ formation.

Concerning **M2**, it has been found that the YM210W mutation induces an increase in the $P^+/P$ redox potential such that, according to the Marcus equation, the ET rate is decreased and stabilization of $P^+B_A^-$ is considerably suppressed[27]. Using 2DES, we surprisingly found that charge separation in **M2** did not involve the $P_A^+P_B^-$ intermediate CT state, and direct ET from $P^*$ to $B_A$ was much slower at $(260\,ps)^{-1}$. The reason for the absence of a $P_A^+P_B^-$ state may be that the relative configurations of $P_A$ and $P_B$ change and, as a result, the barrier for charge separation increases.

In **M3**, the ultrafast $P_A^+P_B^-$ intermediate state was still involved in charge separation. Its formation time constant, 190 fs, was about twice the 110 fs recorded for **M1**. The overall time constant of the reversible $P_A^+P_B^- \rightleftharpoons P^+B_A^-$ ET was 25 ps. It indicated that the forward ET was slower than that in **M1**, which may be due to the change of the $P^* \rightarrow P^+B_A^-$ activation energy. However, the more

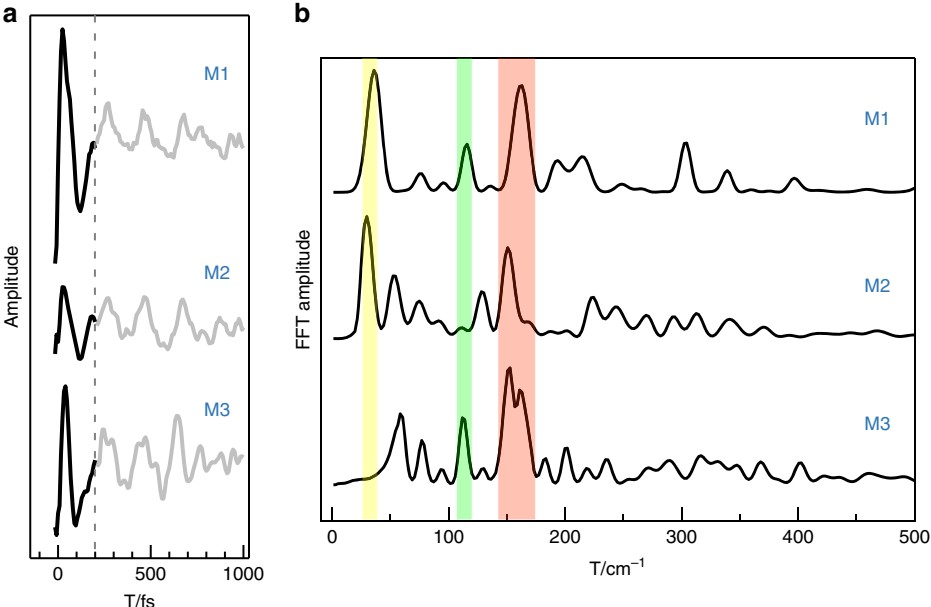

**Fig. 4** Quantum beats and Fourier transform (FT) power spectra. The top, middle, and bottom traces represent **M1**, **M2**, and **M3**. **a** Real-part $T$ traces at the cross-peak locations as labeled with a cross in Fig. 3 after subtraction of multiexponential dynamics. **b** Summary FT power spectra of the oscillations in the real-part signals (0–2 ps)

important reason was the increased efficiency of backward ET, $P_A^+P_B^- \leftarrow P^+B_A^-$. Because this occurred only in **M3**, we conclude that it was the result of structural changes associated with the GM203L mutation, the main effect of which is the removal of a water molecule directly linking $P_B$ and $B_A$[29]. Various experimental and theoretical simulations have studied the role of this water, showing that its presence increased the coupling between P and $B_A$[22], reduced the energy of $P^+B_A^-$ by ~600 cm$^{-1}$, and stabilized $P^+B_A^-$[40,41]. Our results consolidate previous ideas by showing that increased backward ET considerably reduced the stability of $P^+B_A^-$.

**Electronic coherence.** Electronic coherence in pigment–protein complexes can hardly survive >200 fs[12], and hence only $P_A^+P_B^-$ formed on an ultrafast timescale can possibly be coherent with $P^*$. To verify this, we extracted the QB (Fig. 4a) of the $(P^*, P_A^+P_B^-)$ cross peaks (marked by crosses in Fig. 3), by subtracting the multiexponential population dynamics from the experimental $T$ dynamics. This positive below-diagonal cross peak represented coupling and tranfering an electron between $P^*$ and $P_A^+P_B^-$.

In **M1**, the QB was high amplitude at early time and it remained until ~200 fs. The oscillatory period/frequency was approximately 190 fs/193 cm$^{-1}$, comparable with the energy splitting between $P^*$ and $P_A^+P_B^-$ (160–260 cm$^{-1}$, estimated by decomposing the absorption spectrum into $P^*$ being at 884.3 nm and $P_A^+P_B^-$ at 897–905 nm). This high-amplitude rapidly dephasing QB only appeared around the $(P^*, P_A^+P_B^-)$ location; at other locations such as ESA peak or (B, P) cross peak nearly no similar oscillation was observed (see Supplementary Note 4 and Supplementary Fig. 6). It is a strong indication of a coherent superposition of $P^*$ and $P_A^+P_B^-$. Other possibilities, such as environmental effects or variants of ET tunneling matrix, cannot produce this QB. Very recently, a similar QB in the peridinin-chlorophyll $a$-protein was assigned to electronic coherence[42]. The $P^* \rightarrow P_A^+P_B^-$ reaction had a 110-fs time constant that was comparable to the dephasing time, and thus it can be concluded that the initial charge separation was an electronically coherent process. A similar QB was also observed at room temperature for

**M1** (see Supplementary Note 5 and Supplementary Fig. 7), indicating that electronic coherence was also involved in charge separation under physiologically relevant conditions.

In **M3**, a similar electronic coherence with a 199 cm$^{-1}$ frequency was also identified that corresponded to a decrease of the charge separation rate by about two-fold, (190 fs)$^{-1}$. However, the amplitude of QB was somewhat smaller than that in **M1**. We speculate that the strength of coherent interaction, i.e., the coupling between $P^*$ and $P_A^+P_B^-$, was weaker in **M3**, due to the configuration changes of the cofactors. In **M2**, the high-amplitude QB nearly disappeared and it corresponded to the absence of $P_A^+P_B^-$. Taking the data for **M1**, **M3**, and **M2** together, a trend was observed that was consistent with electronic coherence facilitating the $P^* \rightarrow P_A^+P_B^-$ charge separation and playing an important role for the super-high efficiency seen in the native RC (equating to **M1** in our work).

**Vibrational coherence.** The 2D spectra also exhibited long-lived (up to 2 ps) oscillations, which were considered as arising from vibrational/vibronic coherence[12,24]. Summary FT power spectra, obtained by summing the squared absolute value (Frobenius norm) of the FTs of the $T$ traces across the 2D spectra, are compared in Fig. 4b. Summary spectra were used to eliminate undesired oscillations attributable to noise. For the **M1** RC, the main frequencies were at 35, 115 and 162 cm$^{-1}$. All three had equivalents in a 77-K resonance Raman spectrum of P in wild-type RCs[43]. They deviated slightly from our previous room-temperature data of **M1**: 33, 107 (weak), and 153 cm$^{-1}$[24], as a consequence of the conformation change in BChl $a$ upon freezing[43]. Analysis of the 2D frequency maps, the amplitude distribution of a certain $\omega_T$ frequency in the 2D spectra, showed that these oscillations were predominantly vibrational in origin (Supplementary Note 6 and Supplementary Fig. 8). To understand the potential role of these vibrational modes, we compared their characteristics in the three mutant RCs.

**162 cm$^{-1}$.** The strong peak with a 162 cm$^{-1}$ frequency in **M1** down-shifted to 151 cm$^{-1}$ in **M2** and split into two peaks at 151 and 162 cm$^{-1}$ in **M3**. Previous pump–probe studies on mutant RCs have demonstrated that the ~150 cm$^{-1}$ oscillation was

| | $P_A^+P_B^-$ formation | $P^+B_A^-$ stabilization n | Electronic coherence <200 fs | Vibrational coherence ~2 ps | |
|---|---|---|---|---|---|
| | | | ~193 cm$^{-1}$ | 115 cm$^{-1}$ | 35 cm$^{-1}$ |
| **M1** | ✓ | ✓ | ✓ | ✓ | ✓ |
| **M2** | ✗ | ✓ | ✗ | ✗ | ✓ |
| **M3** | ✓ | ✗ | ✓ | ✓ | ✗ |

*RC* reaction center

**Table 1 The behavior of the three mutant RCs**

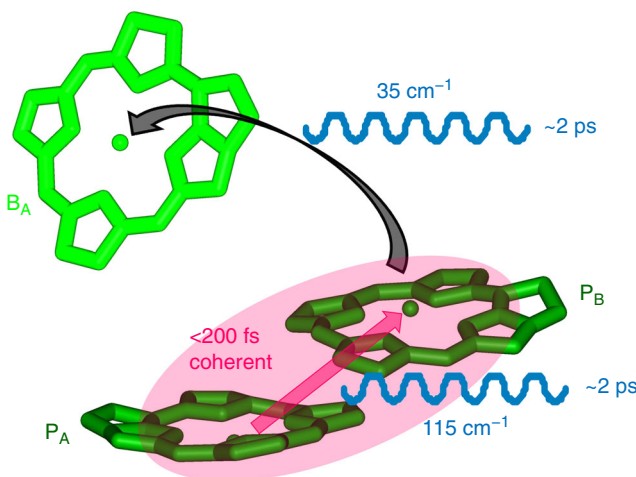

**Fig. 5** Coherent primary electron transfer. A scheme of the primary electron transfer process facilitated by coherences

activated inside the P BChl dimer and was not abolished by either the YM210W and GM203L mutation, although its frequencies shifted somewhat within the 140−170 cm$^{-1}$ region[19,27,30,31,41]. Our 2DES study supported this observation. Because it did not exhibit large changes when the charge separation process changed substantially, we concluded that this vibrational mode may not represent the reaction coordinate of charge separation in RC.

**115 cm$^{-1}$.** This vibrational mode in **M1** was conserved in **M2** but disappeared in **M3**. Theoretical studies have established that the ~115 cm$^{-1}$ mode (~130 cm$^{-1}$ in ref. [20]) determines the initial reaction coordinate of P*, its coupling with the intermolecular dynamics within the P dimer effectively removing the barrier for charge separation, initiating a directional displacement of electron density that leads to the $P_A^+P_B^-$ CT state[20,21]. The absence of the 115 cm$^{-1}$ mode in **M2** was correlated with the fact that no $P_A^+P_B^-$ was formed, which suggested that the coordinate of charge separation was along this mode. We concluded that this vibrational coherence with a 115 cm$^{-1}$ frequency plays a role for charge separation process being rapid.

**35 cm$^{-1}$.** The peak with a 35 cm$^{-1}$ frequency for **M1** and 30 cm$^{-1}$ frequency for **M2** had large amplitudes but was nearly absent in **M3**. Previous experiments have found that the 30−35 cm$^{-1}$ vibrational mode disappears when the water molecule between $P_B$ and $B_A$ was removed[3,31,41]. Theoretical studies identified this mode to be a proton displacement of this water that stabilizes $P^+B_A^-$ by reducing the driving force for charge recombination[22,40]. From 2D-EAS analysis, we found that the backward reaction of $P_A^+P_B^- \rightleftharpoons P^+B_A^-$ ET was much more effective in **M3** than in **M1** or **M2**. Correlating that fact with the present finding that the 30

−35 cm$^{-1}$ mode was lost in **M3**, we can conclude that this vibrational coherence is essential for the directionality of the primary ET process.

## Discussion

It has long been believed that low-frequency coherent nuclear motions facilitate ET processes. In 2014, our and Ogilvie's groups reported that electron/vibronic coherence may play a role for enhancing charge separation in the PSII RC[8,9]. However, later with the evolution of assignments of the slowly dephasing QBs, diverse coherences were identified with different dephasing lifetimes. For example, ~100 fs electronic coherence and 2 ps vibrational coherence were distinguished in the Fenna–Matthews–Olsen complex[12], and it was suggested that the electronic coherence did not contribute to that energy transfer process. Very recently, a combined quantum dynamic–chemical calculation advanced that individual vibrational modes play a minor role in promoting charge separation but collectively complement the robustness of the charge separation[44]. Hence, it is important to reformulate the mechanism behind the highly efficient solar-energy conversion in RCs. This work entails a detailed 2DES study on three mutant RCs with different primary ET rates achieved through well-characterized changes to the structure of the protein–cofactor matrix. By comparing dynamics, QBs, and ET efficiencies, we have successfully identified one electronic and two vibrational coherences and suggested their respective roles in the primary ET process. An overall scheme is shown in Table 1 and Fig. 5.

In mutant **M1**, whose $P^* \rightarrow P^+H_A^-$ ET process is nearly identical with that in the wild-type RC, the primary ET process consists of two steps, $P^* \xrightarrow{110\,fs} P_A^+P_B^- \underset{1.4\,ps}{\rightleftharpoons} P^+B_A^-$. In this scheme, formation of the internal CT state $P_A^+P_B^-$ can be viewed as a precursor step to the first ET. We can see that the overall rate of the primary ET, 1.4 ps, is not as fast as other processes such as energy transfer from B to P (200 fs). If the precursor step was not involved, a high efficiency cannot be ensured. Thus the essential factor for the high efficiency is the formation of the $P_A^+P_B^-$ intermediate.

We found that two conditions are associated with $P_A^+P_B^-$ formation, an electronic coherence between P* and $P_A^+P_B^-$ and a vibrational coherence leading to directional displacement of electron density. The electronic coherence gave a high-amplitude QB before ~200 fs that was distinguishable from more long-lived vibrational coherence. We observed a direct correlation between $P_A^+P_B^-$ formation and the presence of electronic coherence between P* and $P_A^+P_B^-$, strongly suggesting that the initial charge separation is an electronically coherent process. The vibrational coherence with a 115 cm$^{-1}$ frequency also correlated with the $P_A^+P_B^-$ formation, and this vibrational mode may represent the reaction coordinate.

Our group has previously found in the PSII RC that exciton–CT mixing can be dramatically enhanced in the presence of an intramolecular vibrational quantum with an energy that is close to the energy gap between the exciton and

CT states[8,45]. This resonant vibration mechanism may also apply to the *Rba. sphaeroides* RC, considering that the vibrational energy, 115 cm$^{-1}$, was quasi similar to the energy gap of 160−260 cm$^{-1}$ and it appeared or disappeared synchronously with the electronic coherence. We suggest that the electronic and resonant vibrational coherence together lead P* to transit into $P_A^+P_B^-$ with unity efficiency.

In the second step, $P_A^+P_B^- \underset{}{\overset{1.4\,ps}{\rightleftharpoons}} P^+B_A^-$, stabilization of the formed $P^+B_A^-$ is another factor that influences the overall primary ET efficiency: $P^+B_A^-$ is stabilized by a proton displacement involving a vibrational mode of 30−35 cm$^{-1}$[12,40], without which charge recombination would occur. We found that this vibrational coherence was as long-lived as ~2 ps, and its disappearance correlated with the enhanced backward reaction, i.e., more charge recombination. The lifetime, ~2 ps, is comparable to the time constant of this ET step (1.4 ps at 77 K and 3.4 ps at room temperature), ensuring an effective stabilization of $P^+B_A^-$.

The protein matrix not only holds the pigment molecules but also is essential for keeping the coherences and fulfilling the physiological function. For example, in RC, it tightly packs the two P dimer molecules, which cannot be achieved in other environments such as in solution, as a result, P possesses an intradimer CT character. This is the basis for the coherent superposition of electronic states, P* and $P_A^+P_B^-$. Furthermore, the protein matrix could prolong the vibrational coherences, by synchronously coupling nuclear motions of distinct excitons[46,47].

On the basis of the correlations we found between the ET rates and the QBs with specific frequencies, we suggested that individual vibrational coherences could contribute to primary ET. This idea is supported by several theoretical works[19–22]. However, it is noteworthy that theoretical calculations employing different models and assumptions may arrive at opposite conclusions. For example, by considering not only nuclear degrees of freedom but also protein fluctuations, the nonequilibrium vibrational coherences was found not to contribute very much to accelerate the primary ET rate[44,48,49].

In summary, we propose that the primary ET process in the purple bacterial RC is collectively optimized by electronic and vibrational coherences. Very recently, researchers have begun to study enhancement of the efficiency of solar conversion through the use of materials that exploit coherence phenomena[50]. We believe that the delicate quantum coherence mechanism in the RC, which must be the result of natural evolution during billions of years, could provide guidance for new artificial energy conversion systems that exploit coherence phenomena to achieve maximum efficiency.

## Methods

**Sample preparation.** Single (AM260W) and double (AM260W/YM210W or AM260W/GM203L) RCs modified with a poly-histidine tag on the PufM polypeptide[51] were isolated from a strain *Rba. sphaeroides* lacking light-harvesting complexes[52] using n-dodecyl-N,N-dimethylamine-N-oxide (LDAO) as the solubilizing detergent. RCs were purified by nickel affinity chromatography followed by size exclusion chromatography, as described in detail previously[51]. Samples were diluted with a buffer containing 20 mM Tris (pH 8.0)/0.05% LDAO (w/v) to an optical density of ~0.3 at the maximum of P band for the 2DES measurement in 0.1 mm quartz cell. For the 77 K measurement, 60% glycerol (v/v) was added and the sample was held in a nitrogen-flow cryostat. After each measurement, the degree of the sample degradation was monitored by ultraviolet–visible spectroscopy and <2% degradation was observed.

**Experiment.** 77 K 2DES was carried out with a diffractive optic-based inherently phase-stabilized four-wave mixing set-up, which was described in detail previously[53]. The repetition rate of the laser system (PHAROS, Light Conversion) was set to 1 kHz. The pulse generated by a home-built non-collinear optical parametric amplifier was centered at 880 nm with a full-width half-maximum of 105 nm and a duration of 17 fs. The excitation intensity was 1.5 nJ per pulse, corresponding to a photon density of $2 \times 10^{13}$ photons cm$^{-2}$ at which annihilation was negligible. The

polarization scheme was all parallel. The coherence time $\tau$ was scanned from −90 fs to 100 fs, with 1-fs step by employing movable fused silica wedges. The population time $T$ was scanned from −1 ps to 1 ns, with a step of 8 fs in the −24 fs to 2 ps region. With this laser condition, no before-zero signal was observed, showing that there was no accumulation of long-lived CT components and that the RCs were functionally active.

## Data availability

All data supporting the findings of this study are available from the corresponding author upon request.

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

## Acknowledgements

F.M., E.R. and R.v.G. were supported by an Advanced Investigator grant from the European Research Council (No. 267333, PHOTPROT) to R.v.G and the TOP-grant (700.58.305) from the Foundation of Chemical Science part of NWO. R.v.G. gratefully acknowledges his Academy Professor grant from the Netherlands Royal Academy of Sciences (KNAW). M.R.J. acknowledges support from the Biotechnology and Biological Sciences Research Council of the UK (project BB/I022570/1). V.I.N. was supported by the Russian Foundation for Basic Research (Grant No. 18-04-00105).

## Author contributions

F.M., V.I.N. and R.v.G. designed the research; F.M. and E.R. performed the experiments; M.R.J. constructed the engineered RCs and supplied purified proteins; F.M. analyzed the data and wrote the manuscript; and all authors reviewed the manuscript.
