## [Peer Review File · Nature Communications]

Reviewers' comments:

Reviewer #1 (Remarks to the Author):

In this paper titled "Both electronic and long-lived vibrational coherence contribute to primary electron transfer in the purple bacterial reaction center," the authors have performed two-dimensional electronic spectroscopy (2DES) experiment to investigate primary electron transfer dynamics in three structurally-modified reaction center protein of the purple bacterium *Rhodobacter (Rba.) sphaeroides*. On the basis of the experimental data, the authors have discussed correlations between primary electron transfer rates and quantum beats of electronic and vibrational origins.

The authors addressed the following three mutations of the *Rba. sphaeroides* reaction center:

- AM260W: The alanine at position 260 of the M-polypeptide is replaced by tryptophan. In this reaction center, there exists no oxidized P in the timescale of the 2DES experiment, enabling the authors to investigate the photoinduced primary electron transfer dynamics. Also, the electron transfer rates are almost identical to those in the wild-type reaction center.
- YM210W: The tyrosine at position 210 of the M-polypeptide is replaced by tryptophan. This mutation causes a 50 mV increase in the P+/P mid-point redox potential. Consequently, the primary electron transfer is slowed by as much as two orders of magnitude.
- GM203L: The glycine at position 203 of the M-polypeptide is replaced by leucine. This mutation removes a water molecule linking PB and BA by hydrogen-bond interaction. Consequently, the primary electron transfer is slowed by one-order magnitude.

For the 2DES experiment, the authors discussed

- M1: AM260W ... the electron transfer rates are almost identical to those in the wild-type reaction center.
- M2: AM260W + YM210W ... the electron transfer rates are two-order magnitudes slow in comparison to those in the wild-type reaction center.
- M3: AM260W + FM203L ... the electron transfer rates are one-order magnitude slow in comparison to those in the wild-type reaction center,

Consequently, the authors obtained the results in Figure 5, which demonstrates a certain correlation between the primary charge separation rates and the quantum beats of electronic and vibrational origins, and concluded that both electronic and long-lived vibrational coherence contributed to primary electron transfer in the purple bacterial reaction center.

Overall, I found the demonstrated 2DES experiments for the primary electron transfer dynamics in the three structurally-altered reaction center proteins very intriguing and thought-provoking. However, I wonder if there could exist a large gap for logically concluding the causal relationship between the enhancement of the primary electron transfer and the electronic/vibrational coherence. Although I agree to the point that the 2DES experimental data exhibited some correlations between the electron transfer rates and the presence of the quantum beats, I could not be 100% convinced that they were causally connected. Furthermore, I do not think the authors excluded any other physical reasons for increasing or decreasing the primary electron transfer rates in the structurally-different reaction center proteins.

More specific comments are given below:

- 1) The authors assigned the quantum beats of frequency 197cm^{-1} to the electronic coherence between P^* and $PA+PB^-$. As the reason for this assignment, the authors claimed that the electronic energy of $PA+PB^-$ was 900nm . As long as I see Figure 1, however, I am not sure why the electronic energy of $PA+PB^-$ is 900nm . The authors should clarify this issue or refer to appropriate papers, and the authors should add clarifying words to exclude any other possibilities such as environmental effects, variants of electron transfer tunneling matrix and so force.
- 2) The authors claimed that relatively low frequency vibrations such as 35cm^{-1} , 115cm^{-1} and 162cm^{-1} could contribute to the enhancement of the primary electron transfer and the stabilization of the charge separated state. I agree to the point that the 2DES experimental data at 77K exhibited some correlations between the electron transfer rates and the quantum beats with such frequencies; however, precise physical mechanism is totally unclear. In particular, my concern is these frequencies are smaller than the thermal energy (ca. 200cm^{-1}) at physiological temperature. The authors should clarify more precise mechanism of how the vibrational modes cause the enhancement and stability by overcoming the thermal energies at physiological temperatures.
- 3) The authors should refer to Nat. Chem. 6, 766-711 (2014) by Fuller et al. together with Reference 8.
- 4) Regarding the theoretical support for the idea that the vibrational coherence contributed to primary electron transfer, the authors need to be careful about the theoretical models and assumptions. Indeed, Reference 15 demonstrates that nuclear vibrational modes facilitates electron transfer reaction. However, this kind of claims really depend on the employed model and

assumption. For example, And & Sumi (J. Chem. Phys. B, 102, 10991, 2998), Abramavicius & Valkunas (Photosynth. Res. 127, 33, 2016), and Fujihashi et al (J. Phys. Chem. Lett. 9, 4921, 2018) gave the opposite conclusions with considering not only nuclear degrees of freedom but also protein effects. The authors should discuss the less supportive theoretical papers together with the supportive paper such as References 15 – 18.

Reviewer #2 (Remarks to the Author):

This paper presents a study of three mutant purple bacterial reaction centers that exhibit different electron transfer rates using 2D electronic spectroscopy. By comparing the different mutants the authors make strong conclusions about the importance of electronic and vibrational coherence for the high efficiency of the primary energy conversion steps. In principle the idea of correlating the observation of coherence with electron transfer rate in different mutants is reasonable for establishing possible relevance. However, the authors assume that correlation implies causation and come to conclusions that are too far-reaching and not sufficiently supported by the data as detailed below. If their arguments are supported following the more thorough analysis of the data that is described below it is possible that this paper could be considered for publication. However, in this event, the authors would need to be clear that the study only reveals correlations between observations of coherence and electron transfer rates, suggesting the possibility of functional importance. Establishing causation would require holding all other variables constant and turning on or off the specific coherences to assess the effect on electron transfer, a standard that is not met in the current paper.

1) First sentence of the abstract states “Understanding the quantum coherence mechanism behind the near-unity efficiency of primary electron transfer in reaction centers is vital for designing performance-enhanced artificial solar conversion systems to fulfill mankind’s growing demands for energy”. This sentence implies that it is already established that quantum coherence underlies the near-unity efficiency of primary electron transfer in reaction centers. Although there are some theoretical studies that support this idea (as cited in the paper), and experimental papers that report observations of coherence, the importance of coherence for the near-unity efficiency of electron transfer is still very much open for debate. Recent theory papers do not support this hypothesis for the photosystem II reaction center (Ishizaki JPC Letters 2018 just accepted). Although 2D electronic spectroscopy shows coherent processes very clearly, there is a history of incorrect interpretation of the 2D data that has demonstrated the importance of careful consideration of the entire 2D spectrum and comparisons with models to assign electronic, vibrational or vibronic coherence.

2) The authors should discuss the discrepancies between the frequencies they report in the current paper and their recent JPC Letters paper (ref 19 in the text) which studies mutant M1 at room

temperature. In that study they focus on 153 cm⁻¹ and 33 cm⁻¹ modes which they propose to be important for the high efficiency of charge separation. In the current paper at 77K the frequencies they report as being important for charge separation in M1 are now 193 cm⁻¹ (electronic coherence), and 35, 115 and 162 cm⁻¹ (vibrational coherence). Without clearly justifying reasons for the discrepancies, the comparisons between M1 and the other mutants in the current study are undermined since the conclusions rely on frequency shifts of similar magnitude to the discrepancies between the two studies of M1. Are there resonance Raman studies of mutants M2 and M3? If so, how do the reported frequencies compare with these studies?

3) In comparing the coherences in mutants M1-M3 the authors show FT power spectra in Fig 4 of the paper, taken at particular crosspeak positions chosen for their possible involvement in charge separation, as determined from the global analysis. The authors should not rely on comparing FT spectra from single positions in the 2D spectrum, since this has previously led to many misinterpretations of coherences in 2D data. The authors currently show rephasing frequency maps for M1 (Figure S6) but should instead show full 2D frequency maps (with positive and negative frequencies) for all 3 mutants. They should compare the maps to the expected 2D frequency maps for electronic, vibrational and vibronic coherence (see Butkus CPL 2014,2015 etc.). This would provide better justification for their assignments of excited vs ground state coherence, and electronic vs vibrational coherence for the different mutants. They should also indicate on the 2D frequency maps the location of the crosspeaks which they claim to represent coupling and charge transfer between P* and PA+PB⁻ to substantiate the claim that the coherence amplitude is maximized at this location (it doesn't appear that this is true from Figure S6). The authors claim that the electronic coherence frequencies for M1 and M3 are close to matching the energy splitting between P* and PA+PB⁻. They should provide further discussion of how they determined the PA+PB⁻ energy. Is this from the global analysis in the current study? They should discuss these assignments in the context of other work.

4) The authors should discuss the sensitivity of their reported spectra to the fitting of the kinetics. Since mutants M1-M3 display different kinetics, it is a difficult task to be confident in the low frequency spectra (< 200cm⁻¹) which can be very sensitive to the background subtraction. The authors should offer stronger evidence that the reported spectral differences are meaningful. For example, they should show 2D maps of residuals from the global analysis to demonstrate that the fitting is of comparable high quality for all 3 mutants.

5) Although the authors cite some of the relevant 2D spectroscopy work on the purple bacterial reaction center, they do not provide adequate discussion of the other work. For example, the authors should discuss how the frequencies they observe compare with those observed by Westenhoff et al, JACS 2012, Flanagan et al, JPCA 2015 and Ryu et al. JPCB 2014.

6) Previous pump-probe work on mutants showed differences in low frequency coherences, from which it was concluded that the coherences involved the protein to some degree. The authors should discuss their interpretation of the coherences that they identify – to what degree does the protein contribute to the coherence? How do these coherences relate to the higher frequency ones reported in many other systems such as FMO?

**Reply to Reviewer Comments on Nature Communications Manuscript
NCOMMS-18-23496-T**

Title: Both electronic and long-lived vibrational coherences are involved in primary electron transfer in the purple bacterial reaction center

Authors: Fei Ma, Elisabet Romero, Michael R. Jones, Vladimir I. Novoderezhkin, Rienk van Grondelle

We thank all reviewers for their detailed comments that helped us improve the clarity of the presentation in the revised version of our work. We have addressed all points in detail as listed below, and we have modified the manuscript accordingly.

Especially, both the reviewers query the causal relationship, which makes us realized that the coherences we identified reflect their anticipation and possible functional role in the primary electron transfer process, but they are not necessarily causally connected. Therefore, in the revised manuscript, we modified the title, abstract and several places. At the same time, we carried out a more thorough analysis of the data, to better support the conclusions. For details please see below.

Reviewer #1 (Remarks to the Author):

Overall, I found the demonstrated 2DES experiments for the primary electron transfer dynamics in the three structurally-altered reaction center proteins very intriguing and thought-provoking. However, I wonder if there could exist a large gap for logically concluding the causal relationship between the enhancement of the primary electron transfer and the electronic/vibrational coherence. Although I agree to the point that the 2DES experimental data exhibited some correlations between the electron transfer rates and the presence of the quantum beats, I could not be 100% convinced that they were causally connected. Furthermore, I do not think the authors excluded any other physical reasons for increasing or decreasing the primary electron transfer rates in the structurally-different reaction center proteins.

More specific comments are given below:

1) The authors assigned the quantum beats of frequency 197cm^{-1} to the electronic coherence between P^* and $PA+PB^-$. As the reason for this assignment, the authors claimed that the electronic energy of $PA+PB^-$ was 900nm . As long as I see Figure 1, however, I am not sure why the electronic energy of $PA+PB^-$ is 900nm . The authors should clarify this issue or refer to appropriate papers, and the authors should add clarifying words to exclude any other possibilities such as environmental effects, variants of electron transfer tunneling matrix and so force.

Reply: We thank the reviewer for pointing out this possibly misleading energy of $P_A^+P_B^-$.

Actually 900 nm is just an estimated value and we cannot determine the accurate value.

We estimated it by decomposing the P band with two Gaussian peaks and obtained the

red-tail that correspond to $P_A^+P_B^-$. However, this fitting result is not very reliable, due to the irremovable contribution from other species such as B or background. Each value in 897-905 nm region can give an acceptable fit. Thus, in the revised manuscript, we changed the peak wavelength to 897-905 nm to avoid significant error. As a result, the energy splitting between P^* and $P_A^+P_B^-$ was change to 160-260 cm^{-1} . The changes were marked in red words in **Line 1-2 Page 18**.

Previous calculation of P has determined the lowest energy of P (which is mixed with $P_A^+P_B^-$) as 890 nm at 298 K [Jordanides et al, JPCB, 2001, 105:1652-1669], The energy splitting between P^* and $P_A^+P_B^-$ was thus 250 cm^{-1} , in the 160-260 cm^{-1} region we estimated.

One of the main reasons we assign the electronic coherence is that the high-amplitude short-lived oscillation only appeared around the (P^* , $P_A^+P_B^-$) position, at other locations such as ESA peak or (P, B) cross peak no similar oscillation was observed. Very recently, a similar high-amplitude short-lived oscillation at cross peak was assigned to electronic coherent energy transfer from carotenoid to chlorophyll (has been added as **Reference 39**). We think this strong evidence could help us to exclude other possibilities. And, to our knowledge, environmental effects or variants of tunneling matrix cannot produce this oscillation. A clarifying sentence has been added in **the first paragraph Page 18**.

2) The authors claimed that relatively low frequency vibrations such as 35 cm^{-1} , 115 cm^{-1} and 162 cm^{-1} could contribute to the enhancement of the primary electron transfer and the stabilization of the charge separated state. I agree to the point that the 2DES experimental data at 77K exhibited some correlations between the electron transfer rates and the

quantum beats with such frequencies; however, precise physical mechanism is totally unclear. In particular, my concern is these frequencies are smaller than the thermal energy (ca. 200cm⁻¹) at physiological temperature. The authors should clarify more precise mechanism of how the vibrational modes cause the enhancement and stability by overcoming the thermal energies at physiological temperatures.

Reply: The $P^* \rightarrow P^+B_A^-$ primary electron transfer consists of two steps: $P^* \rightarrow P_A^+P_B^-$ and $P_A^+P_B^- \rightarrow P^+B_A^-$. The $P^* \rightarrow P_A^+P_B^-$ step occurs inside the P dimer, and due to the intra-dimer CT character, the P and $P_A^+P_B^-$ states are coherent superposition. Thus, this step is an electronic coherent process and is independent on temperature (~100 fs for both 77 K and RT). In this step, P* couples the 115 cm⁻¹ mode, lowering the P_B dimer half energy, removing the barrier for charge separation and leading to directional displacement of electron density [Reference 17]. This step determines the unidirection and high efficiency of the whole primary electron transfer process.

The $P_A^+P_B^- \rightarrow P^+B_A^-$ step is an electron transfer process. It involves a proton displacement with a 35 cm⁻¹ vibrational mode [Reference 18]. This proton displacement stabilizes the $P^+B_A^-$ state, however, it does not need to overcome the thermal energy. This step, actually, is temperature dependent (the overall primary electron transfer time constant is 1.4 ps at 77 K while 3.4 ps for RT). We think the 35 cm⁻¹ mode just functions in stabilizing the $P^+B_A^-$ state, but does not enhance the electron transfer rate.

3) The authors should refer to Nat. Chem. 6, 766-711 (2014) by Fuller et al. together with Reference 8.

Reply: This paper has been added as **Reference 9**, and related introduction was added.

4) Regarding the theoretical support for the idea that the vibrational coherence contributed to primary electron transfer, the authors need to be careful about the theoretical models and assumptions. Indeed, Reference 15 demonstrates that nuclear vibrational modes facilitates electron transfer reaction. However, this kind of claims really depend on the employed model and assumption. For example, And & Sumi (J. Chem. Phys. B, 102, 10991, 1998), Abramavicius & Valkunas (Photosynth. Res. 127, 33, 2016), and Fujihashi et al (J. Phys. Chem. Lett. 9, 4921, 2018) gave the opposite conclusions with considering not only nuclear degrees of freedom but also protein effects. The authors should discuss the less supportive theoretical papers together with the supportive paper such as References 15 – 18.

Reply: We agree with the reviewer that there are various theoretical models, and some of them suggest different mechanisms. Thus, a discussion has been added in **the second paragraph in Page 24**, and these papers were cited as **References 41, 44 and 45**.

Reviewer #2 (Remarks to the Author):

This paper presents a study of three mutant purple bacterial reaction centers that exhibit different electron transfer rates using 2D electronic spectroscopy. By comparing the different mutants the authors make strong conclusions about the importance of electronic and vibrational coherence for the high efficiency of the primary energy conversion steps. In principle the idea of correlating the observation of coherence with electron transfer rate in different mutants is reasonable for establishing possible relevance. However, the authors assume that correlation implies causation and come to conclusions that are too

far-reaching and not sufficiently supported by the data as detailed below. If their arguments are supported following the more thorough analysis of the data that is described below it is possible that this paper could be considered for publication. However, in this event, the authors would need to be clear that the study only reveals correlations between observations of coherence and electron transfer rates, suggesting the possibility of functional importance. Establishing causation would require holding all other variables constant and turning on or off the specific coherences to assess the effect on electron transfer, a standard that is not met in the current paper.

1) First sentence of the abstract states “Understanding the quantum coherence mechanism behind the near-unity efficiency of primary electron transfer in reaction centers is vital for designing performance-enhanced artificial solar conversion systems to fulfill mankind’s growing demands for energy”. This sentence implies that it is already established that quantum coherence underlies the near-unity efficiency of primary electron transfer in reaction centers. Although there are some theoretical studies that support this idea (as cited in the paper), and experimental papers that report observations of coherence, the importance of coherence for the near-unity efficiency of electron transfer is still very much open for debate. Recent theory papers do not support this hypothesis for the photosystem II reaction center (Ishizaki JPC Letters 2018 just accepted). Although 2D electronic spectroscopy shows coherent processes very clearly, there is a history of incorrect interpretation of the 2D data that has demonstrated the importance of careful consideration of the entire 2D spectrum and comparisons with models to assign electronic, vibrational or vibronic coherence.

Reply: This sentence does not imply that it is already established that quantum coherence underlies the near-unity efficiency of primary electron transfer in reaction centers. This sentence implies that it is vital to understand these phenomena, which Ishizaki agrees with. This is not the place to explain the problems associated with Ishizaki's work, instead, in **Page 21**, we have added a short description of Ishizaki's work and cited it as **Reference 41**. Here we will replace 'vital' by 'essential'.

There indeed is a history of incorrect interpretation of the 2D data, as pointed in Reference 11 and 12. Our work is just trying to identify and distinguish different coherences, being benefited from comparing different mutants.

2) The authors should discuss the discrepancies between the frequencies they report in the current paper and their recent JPC Letters paper (ref 19 in the text) which studies mutant M1 at room temperature. In that study they focus on 153 cm⁻¹ and 33 cm⁻¹ modes which they propose to be important for the high efficiency of charge separation. In the current paper at 77K the frequencies they report as being important for charge separation in M1 are now 193 cm⁻¹ (electronic coherence), and 35, 115 and 162 cm⁻¹ (vibrational coherence). Without clearly justifying reasons for the discrepancies, the comparisons between M1 and the other mutants in the current study are undermined since the conclusions rely on frequency shifts of similar magnitude to the discrepancies between the two studies of M1. Are there resonance Raman studies of mutants M2 and M3? If so, how do the reported frequencies compare with these studies?

Reply: A comparison of the Raman spectra of P at 278 and 95 K showed that freezing

RC could induce shift of Raman bands and change of their relative amplitudes [Reference 40]. For example, (1) the 145 cm⁻¹ band disappeared at 95 K, instead, a new band at 168 cm⁻¹ appear: (2) the 96 cm⁻¹ band at 278 K shifted to 101 cm⁻¹, and the latter became much more pronounced. The main reason, as we speculate, may be the change in BChl *a* conformation. It is very possible, because for the low-temperature measurements, glycerol is usually added. Glycerol molecules fill in the protein matrix and consequently could to some extent modify the conformation of BChl *a*. A temperature-dependent Raman spectroscopic study proved this speculation [Ivancich et al, Biochemistry, 1997]. Thus, it is not strange that we also have these discrepancies: 35, 115 and 162 cm⁻¹ at 77 K compared to 33, 107 (weak) and 153 cm⁻¹ at room temperature. For the three mutants, they were frozen with the same way, as a result, they could experience similar structural influences and this kind of discrepancy is not expected between them. Thus, it is safe to compare their FT spectra. Furthermore, we actually do not rely on frequency shifts to arrive to our conclusion. Instead, the main evidence is the almost disappearance of the 115 cm⁻¹ band in **M2** and the disappearance of the 35 cm⁻¹ band in **M3** (only very weak amplitude at each position still remains). The 193 cm⁻¹ was obtained by estimating the period of the strong oscillation at early time, not from the FT spectra.

We did not find reports of resonance Raman studies of mutants **M2** and **M3**. However, previously other groups reported the FT spectra of the oscillatory kinetics obtained from pump-probe measurements, which are consistent with our findings: (1) the 94 cm⁻¹ band (corresponding to the 115 cm⁻¹ one in our work) in wild-type RC became much weaker in the YM210W RC [Reference 23]; (2) the 32 cm⁻¹ band (corresponding to the 35 cm⁻¹ one

in our work) in wild-type RC disappeared in the GM203L RC [Reference 27].

3) In comparing the coherences in mutants M1-M3 the authors show FT power spectra in Fig 4 of the paper, taken at particular crosspeak positions chosen for their possible involvement in charge separation, as determined from the global analysis. The authors should not rely on comparing FT spectra from single positions in the 2D spectrum, since this has previously led to many misinterpretations of coherences in 2D data. The authors currently show rephasing frequency maps for M1 (Figure S6) but should instead show full 2D frequency maps (with positive and negative frequencies) for all 3 mutants. They should compare the maps to the expected 2D frequency maps for electronic, vibrational and vibronic coherence (see Butkus CPL 2014,2015 etc.). This would provide better justification for their assignments of excited vs ground state coherence, and electronic vs vibrational coherence for the different mutants. They should also indicate on the 2D frequency maps the location of the cross peaks which they claim to represent coupling and charge transfer between P^* and $PA+PB^-$ to substantiate the claim that the coherence amplitude is maximized at this location (it doesn't appear that this is true from Figure S6). The authors claim that the electronic coherence frequencies for M1 and M3 are close to matching the energy splitting between P^* and $PA+PB^-$. They should provide further discussion of how they determined the $PA+PB^-$ energy. Is this from the global analysis in the current study? They should discuss these assignments in the context of other work.

Reply: We agree that comparing the full 2D frequency maps for the 3 mutants is the more reliable way to distinguish different coherences, and hence have changed **Figure S7**

by the $-\omega_T$ and $+\omega_T$ frequency maps. A detailed analysis was added in **Supplementary Section 7**. The main conclusion from these maps is that the long-lived oscillations originated from vibrational coherence. For some modes we could determine their excited-state nature, while for the others we cannot due to the complex shapes.

We claimed that the coherence amplitude of superposition between P^* and $P_A^+P_B^-$ was maximized at the $(P^*, P_A^+P_B^-)$ location, because at other locations such as ESA peak or (P, B) cross peak, the high-amplitude short-lived oscillation did not appear. However, it can hardly be identified from frequency maps, because the FT spectra and frequency maps were Fourier transformed from the $\omega_T = 0-2$ ps data where the long-lived oscillations were the major components, while the electronic coherence lasted for only ~ 200 fs and was the minor component. Only in the -115 cm^{-1} map of **M1**, the $(P^*, P_A^+P_B^-)$ cross peak appeared. It is impossible to obtain an accurate frequency map of the pure electronic by Fourier transform of the $\omega_T = 0-200$ fs data, due to the subcycle dephasing times. Very recently, a similar high-amplitude short-lived oscillation at cross peak was assigned to electronic coherent energy transfer from carotenoid to chlorophyll (has been added as **Reference 39**).

We estimated the $P_A^+P_B^-$ energy by decomposing the P band with two Gaussian peaks and obtained the red-tail that correspond to $P_A^+P_B^-$. However, this fitting result is not very reliable, due to the irremovable contribution from other species. Each value in 897-905 nm region can give an acceptable fit. Thus, the energy splitting between P^* and $P_A^+P_B^-$ was estimated in the 160-260 cm^{-1} region. We have changed these values, which were marked in red words in **Line 1-2 Page 18**. Previous calculation of P has determined the

lowest energy of P (which is mixed with $P_A^+P_B^-$) as 890 nm at 298 K [Jordanides et al, JPCB, 2001, 105:1652-1669], The energy splitting between P^* and $P_A^+P_B^-$ was thus 250 cm^{-1} , in the 160-260 cm^{-1} region we estimated.

4) The authors should discuss the sensitivity of their reported spectra to the fitting of the kinetics. Since mutants M1-M3 display different kinetics, it is a difficult task to be confident in the low frequency spectra ($< 200cm^{-1}$) which can be very sensitive to the background subtraction. The authors should offer stronger evidence that the reported spectral differences are meaningful. For example, they should show 2D maps of residuals from the global analysis to demonstrate that the fitting is of comparable high quality for all 3 mutants.

Reply: We agree that the low frequency spectra can be very sensitive to the background subtraction and have added the 2D maps of fitting residuals as **Fig. S2**. They show comparable relative amplitudes and similar evolution trends between the three mutants.

5) Although the authors cite some of the relevant 2D spectroscopy work on the purple bacterial reaction center, they do not provide adequate discussion of the other work. For example, the authors should discuss how the frequencies they observe compare with those observed by Westenhoff et al, JACS 2012, Flanagan et al, JPCA 2015 and Ryu et al. JPCB 2014.

Reply: In those works, most of the RCs under investigation have oxidized P and cannot undergo charge separation. Furthermore, the focus of them are the vibronic mechanisms associated with the energy transfer from H to B, not the charge separation from P to B.

For example, in Westenhoff et al, JACS 2012 and Ryu et al. JPCB 2014, P were oxidized and the excitation pulse even did not cover P. In Flanagan et al, JPCA 2015, although P was active and the excitation pulse covered all B, H and P, the signal amplitude of P was so small that accurate oscillatory component can hardly be derived. The FT spectra thus reflected the vibrational modes belonging to B and H. Similar work was the Reference 11, where the 560 and 650 cm^{-1} modes were found nearly resonant with B-H and facilitate the energy transfer.

Above all, the FT frequencies in those works are the modes of B and H, while what we focused in the present work are the modes of P. Therefore, it is meaningless to compare them directly.

6) Previous pump-probe work on mutants showed differences in low frequency coherences, from which it was concluded that the coherences involved the protein to some degree. The authors should discuss their interpretation of the coherences that they identify – to what degree does the protein contribute to the coherence? How do these coherences relate to the higher frequency ones reported in many other systems such as FMO?

Reply: We think that the protein holds the pigment molecules and is essential for keeping the coherences. Firstly, it tightly packs the two P dimer molecules, which cannot be achieved in other environments such as in solution, as a result, P possesses intra-dimer CT character. This is the basis for the coherent superposition of electronic states, P^* and $P_A^+P_B^-$. Secondly, as advanced in a recent paper [has been added as

Reference 42], the protein plays an important role for prolonging the vibrational coherences, by synchronously coupling nuclear motions of distinct excitons. A short discussion about the role of protein on the coherences has been added in **the last paragraph in Page 23**.

We think that B and P in RC are the more suitable system to compare with, rather than FMO complex. In this work, the FT spectra predominantly consist of low-frequency (<350 cm⁻¹) peaks and reflect the vibrational modes of P. Whereas in the FT spectra representing H→B energy transfer, the main peaks are 500–900 cm⁻¹ [Westenhoff et al, JACS 2012; Reference 11]. We speculate that for different functions, energy transfer or electron transfer, a system employs different vibrational modes to fulfill. One determining factor could be energy resonance, i.e. the vibrational frequency matches the energy difference between electronic states. For example, the 560 and 650 cm⁻¹ matches the energy gap between H and B; while the 193 and 115 cm⁻¹ match the energy gap between P and $P_A^+P_B^-$.

Reviewers' comments:

Reviewer #1 (Remarks to the Author):

I think that the authors have addressed all of my comments in a satisfactory fashion.

However, because the authors have not clearly demonstrated the causal relationship between the experimentally observed electronic/vibrational coherence and the enhancement of the primary charge separation, I am not so sure to what extent this work is novel and prominent in comparison to the preceding studies. Also, I think that the added words in the abstract "suggest that they may play an important role for achieving a super-high efficiency" would be misleading, and therefore I would like to suggest the authors delete them.

Reviewer #2 (Remarks to the Author):

The authors have acknowledged that their claim of a causal relation between the presence of coherence and the electron transfer rate should be more appropriately described as a correlation. They have made several changes in the manuscript to reflect this. However, there remain many other places in the manuscript that maintain the language of the original submission and overstate the claims of the study as detailed below. Some additional revisions are also required to justify the current claims of the study.

In their response to comment 1) in my review, the authors claim that the following sentence "does not imply that it is already established that quantum coherence underlies the near-unity efficiency of primary electron transfer in reaction centers". I repeat the sentence in question here (line 22 of the paper): "Understanding the quantum coherence mechanism behind the near-unity efficiency of primary electron transfer in reaction centers is vital for designing performance-enhanced artificial solar conversion systems to fulfill mankind's growing demands for energy". The term "quantum coherence mechanism" clearly attributes quantum coherence to the mechanism. They should instead remove "quantum coherence" and instead write "Understanding the mechanism behind the near-unity efficiency of primary electron transfer in reaction centers is vital for designing performance-enhanced artificial solar conversion systems to fulfill mankind's growing demands for energy".

line 466: “In summary, we have shown that the primary electron transfer process in the purple bacterial RC is collectively optimized by electronic and vibrational coherence”. The authors must remove “we have shown” and replace it with a more appropriate term such as “we propose” or “we suggest”.

In their response to comment 2) in my review, the authors argue that solvent and temperature differences may explain the differences in coherence frequencies between their previous study and the current work. They should include some discussion of these differences and their explanation for the differences in the manuscript rather than exclusively in the response letter.

The authors now include the positive and negative frequency maps in Figure S7 and provide some discussion of the maps, mainly in the SI. The authors claim that “The absence of the 115 cm⁻¹ mode in M2 was correlated with the fact that no $[[P_A^+ P]]_B^-$ was formed, which directly proved that the coordinate of charge separation was along this mode. It can be concluded that this vibrational coherence with a 115 cm⁻¹ frequency is essential for the charge separation process to be rapid.” Once again these claims are considerably too strong. The authors should mark the cross-peak location (shown in Fig 3a) that they attribute to the PA+PB- CT state on Figure S7 and discuss how the maps support their claim that the 115cm-1 mode is essential. While there appears to be some amplitude at the cross-peak position in the positive frequency 115 cm-1 map, the maximum coherence is considerably shifted from this position, which is not consistent with the authors’ argument. There is considerably more amplitude at that location in the negative frequency map, corresponding to ground state coherence. The authors should discuss this discrepancy and modify the language of the quoted text above to more appropriately reflect the speculative nature of their claim.

The authors correctly point out that the frequency maps cannot report on the electronic coherence due to its short-lived nature. However, the analysis presented in Figure 4 is not sufficiently rigorous to support their claims. In new text added on line 333 they claim that the high amplitude rapidly-dephasing quantum beat only appears at the (P*,PA+PB-) location. They must substantiate this claim by showing more than a single T trace for the M1 mutant in Figure 4a. Instead of a single trace, they should plot the 2D distribution of the amplitude of the quantum beat feature, in analogy to a frequency map, and show that the maximum amplitude appears at the purported (P*,PA+PB-) location as claimed.

On line 446: “We found that two conditions are necessary for $[[P_A^+ P]]_B^-$ formation, an electronic coherence between P* and $[[P_A^+ P]]_B^-$ and a vibrational coherence leading to directional displacement of electron density”. The authors should replace “necessary” with a more appropriate claim.

On line 430: “We observed a direct correlation between $[[P_{A^+} P]]_{B^-}$ formation and the presence of electronic coherence between P^* and $[[P_{A^+} P]]_{B^-}$, strongly proving that the initial charge separation is an electronically coherent process”. The authors should replace “proving” with “suggesting”.

In the discussion of Figure S7 the authors include the statement (page 11) “In the main text we have showed that both the 115 cm^{-1} vibrational mode and the electronic coherence between P^* and $[[P_{A^+} P]]_{B^-}$ were necessary for the formation of $[[P_{A^+} P]]_{B^-}$.” This statement is not sufficiently supported by the analysis and should be modified.

On line 456 the authors cite the recent work by Rolczynski et al. that claims the protein matrix couples the nuclear motions of distinct excitons. The authors should also cite other work that counters this claim, such as Obrich et al, JPC Letters, 2011, 2, 1771.

The authors point out differences between their work and previous 2D spectroscopy studies of the purple bacterial RC as justification for not citing the other work. This gives the impression that they are the only group to have reported coherences in this system. Although they correctly point out that most previous work focused on oxidized reaction centers they should nonetheless cite the other studies and provide a brief statement clarifying the differences between the studies. This will help readers find other relevant work.

Finally, there are many places in the manuscript and SI where a thorough proof-read is needed. For example, the uses of “rephrasing” instead of “rephasing” in the SI.

Reply to Reviewer Comments on Nature Communications Manuscript NCOMMS-18-23496A

Title: Both electronic and long-lived vibrational coherences are involved in primary electron transfer in the purple bacterial reaction center

Authors: Fei Ma, Elisabet Romero, Michael R. Jones, Vladimir I. Novoderezhkin, Rienk van Grondelle

We again thank the reviewers for their suggestions and corrections, and we have modified the manuscript to address the requests. Below the point-to-point responses:

Reviewer #1 (Remarks to the Author):

I think that the authors have addressed all of my comments in a satisfactory fashion.

However, because the authors have not clearly demonstrated the causal relationship between the experimentally observed electronic/vibrational coherence and the enhancement of the primary charge separation, I am not so sure to what extent this work is novel and prominent in comparison to the preceding studies. Also, I think that the added words in the abstract "suggest that they may play an important role for achieving a super-high efficiency" would be misleading, and therefore I would like to suggest the authors delete them.

Reply: This experimental work revealed the correlation between the electronic/vibrational coherence and the super-high electron transfer (ET) efficiency. The novel and prominent points are: (1) the 2DES feature of the ET intermediate, $P_A^+P_B^-$, was reported; (2) the ultrafast electronic coherence between P^* and $P_A^+P_B^-$ was identified,

which, to our knowledge, is the first direct observation of electronic coherence in ET process; (4) except the short-lived electronic coherence, two long-lived vibrational coherences and their functions were also distinguished. By comparing different mutant RCs, we revealed how these coherences influenced the ET rates. The results provided a detailed and thorough understand of the coherent mechanism behind the high ET efficiency. We agree that the above-mentioned sentence is misleading, so we modified it into “they correlated with the achievement of a super-high efficiency”, instead of deleting it.

We did not include any theoretical improvement on how electronic/vibrational coherences enhance ET rate, which is far beyond this work. We will study this issue in some next (more special theoretical) paper.

Reviewer #2 (Remarks to the Author):

The authors have acknowledged that their claim of a causal relation between the presence of coherence and the electron transfer rate should be more appropriately described as a correlation. They have made several changes in the manuscript to reflect this. However, there remain many other places in the manuscript that maintain the language of the original submission and overstate the claims of the study as detailed below. Some additional revisions are also required to justify the current claims of the study.

In their response to comment 1) in my review, the authors claim that the following sentence “does not imply that it is already established that quantum coherence underlies the near-unity efficiency of primary electron transfer in reaction centers”. I repeat the sentence in question here (line 22 of the paper): “Understanding the quantum coherence mechanism behind the near-unity efficiency of primary electron transfer in reaction centers is vital for designing performance-enhanced artificial solar conversion systems to fulfill mankind’s growing demands for energy”. The term “quantum coherence mechanism” clearly attributes quantum coherence to the mechanism. They should instead remove “quantum coherence” and instead write “Understanding the mechanism behind the near-unity efficiency of primary electron transfer in reaction centers is vital for designing performance-enhanced artificial solar conversion systems to fulfill mankind’s growing demands for energy”.

Reply: The words “quantum coherence” have been deleted.

line 466: “In summary, we have shown that the primary electron transfer process in the purple bacterial RC is collectively optimized by electronic and vibrational coherence”. The authors must remove “we have shown” and replace it with a more appropriate term such as “we propose” or “we suggest”.

Reply: The words “we have shown” have been replaced with “we proposed”.

In their response to comment 2) in my review, the authors argue that solvent and temperature differences may explain the differences in coherence frequencies between their previous study and the current work. They should include some discussion of these differences and their explanation for the differences in the manuscript rather than exclusively in the response letter.

Reply: A short discussion and explanation has been added in the second paragraph of Page 19.

The authors now include the positive and negative frequency maps in Figure S7 and provide some discussion of the maps, mainly in the SI. The authors claim that “The absence of the 115 cm^{-1} mode in M2 was correlated with the fact that no $P_A^+P_B^-$ was formed, which directly proved that the coordinate of charge separation was along this mode. It can be concluded that this vibrational coherence with a 115 cm^{-1} frequency is essential for the charge separation process to be rapid. ” Once again these claims are considerably too strong.

Reply: This part has been modified as below (in the second paragraph of Page 20):
The absence of the 115 cm^{-1} mode in M2 was correlated with the fact that no $P_A^+P_B^-$ was formed, which suggested that the coordinate of charge separation was along this mode. We concluded that this vibrational coherence with a 115 cm^{-1} frequency plays a role for the charge separation process being rapid.

The authors should mark the cross-peak location (shown in Fig 3a) that they attribute to the $P_A^+P_B^-$ CT state on Figure S7 and discuss how the maps support their claim that the 115 cm^{-1} mode is essential. While there appears to be some amplitude at the cross-peak position in the positive frequency 115 cm^{-1} map, the maximum coherence is considerably shifted from this position, which is not consistent with the authors’ argument. There is considerably more amplitude at that location in the negative frequency map, corresponding to ground state coherence. The authors should discuss this discrepancy and modify the language of the quoted text above to more appropriately reflect the speculative nature of their claim.

Reply: Crosses marking the (P^* , $P_A^+P_B^-$) positions have been added in Figure S7. The

part related with this cross peak has been reformulated in the second paragraph of Page S12. Furthermore, to give a clearer view of relative amplitudes, we changed the displaying way of the Figure S7. It showed that the positive frequency map had more amplitude.

The authors correctly point out that the frequency maps cannot report on the electronic coherence due to its short-lived nature. However, the analysis presented in Figure 4 is not sufficiently rigorous to support their claims. In new text added on line 333 they claim that the high amplitude rapidly-dephasing quantum beat only appears at the (P^* , $P_A^+P_B^-$) location. They must substantiate this claim by showing more than a single T trace for the M1 mutant in Figure 4a. Instead of a single trace, they should plot the 2D distribution of the amplitude of the quantum beat feature, in analogy to a frequency map, and show that the maximum amplitude appears at the purported (P^* , $P_A^+P_B^-$) location as claimed.

Reply: The 2D distribution of amplitude has been added in Supplementary Section 6.

On line 446: “We found that two conditions are necessary for $P_A^+P_B^-$ formation, an electronic coherence between P^* and $P_A^+P_B^-$ and a vibrational coherence leading to directional displacement of electron density”. The authors should replace “necessary” with a more appropriate claim.

Reply: The words “necessary for” have been replaced with “associated with”.

On line 430: “We observed a direct correlation between $P_A^+P_B^-$ formation and the presence of electronic coherence between P^* and $P_A^+P_B^-$, strongly proving that the initial charge separation is an electronically coherent process”. The authors should replace “proving” with “suggesting”.

Reply: The replacement has been made.

In the discussion of Figure S7 the authors include the statement (page 11) “In the main text we have showed that both the 115 cm^{-1} vibrational mode and the electronic coherence between P^* and $P_A^+P_B^-$ were necessary for the formation of $P_A^+P_B^-$ ” This statement is not sufficiently supported by the analysis and should be modified.

Reply: This words “necessary for” have been replaced with “associated with”.

On line 456 the authors cite the recent work by Rolczynski et al. that claims the protein matrix couples the nuclear motions of distinct excitons. The authors should also cite other work that counters this claim, such as Obrich et al, JPC Letters, 2011, 2, 1771.

Reply: It has been added as Ref 47.

The authors point out differences between their work and previous 2D spectroscopy

studies of the purple bacterial RC as justification for not citing the other work. This gives the impression that they are the only group to have reported coherences in this system. Although they correctly point out that most previous work focused on oxidized reaction centers they should nonetheless cite the other studies and provide a brief statement clarifying the differences between the studies. This will help readers find other relevant work.

Reply: A short introduction of the 2DES studies on *Rbs. Sphaeroides* RC has been added in the **first paragraph of Page 4**, including adding the mentioned papers as **Ref 15-17**.

Finally, there are many places in the manuscript and SI where a thorough proof-read is needed. For example, the uses of “rephrasing” instead of “rephasing” in the SI.

Reply: We thank the reviewer for the careful corrections and we have corrected the spelling mistakes.

REVIEWERS' COMMENTS:

Reviewer #2 (Remarks to the Author):

The authors have addressed the majority of the problems with the earlier versions of the manuscript. The abstract of the paper still contains an unsubstantiated claim: the authors cannot use the words "necessary for" on line 34. Instead they should more reasonably say "associated with" to make it clear that causation has not been established in their paper.

**Reply to Reviewer Comments on Nature Communications Manuscript
NCOMMS-18-23496B**

Title: Both electronic and vibrational coherences are involved in primary electron transfer in bacterial reaction center

Authors: Fei Ma, Elisabet Romero, Michael R. Jones, Vladimir I. Novoderezhkin, Rienk van Grondelle

Reviewer #2 (Remarks to the Author):

The authors have addressed the majority of the problems with the earlier versions of the manuscript. The abstract of the paper still contains an unsubstantiated claim: the authors cannot use the words "necessary for" on line 34. Instead they should more reasonably say "associated with" to make it clear that causation has not been established in their paper.

Reply: We again thank the reviewer for this correction. The words "necessary for" have been replaced with "associated with".